# Biodiversity–production feedback effects lead to intensification traps in agricultural landscapes

Alfred Burian ●[1,2] ✉, Claire Kremen ●[3,4,5], James Shyan-Tau Wu[3], Michael Beckmann[1], Mark Bulling[6], Lucas Alejandro Garibaldi ●[7,8], Tamás Krisztin[9], Zia Mehrabi[3,10], Navin Ramankutty ●[3,11] & Ralf Seppelt ●[1,12,13]

Intensive agriculture with high reliance on pesticides and fertilizers constitutes a major strategy for 'feeding the world'. However, such conventional intensification is linked to diminishing returns and can result in 'intensification traps'—production declines triggered by the negative feedback of biodiversity loss at high input levels. Here we developed a novel framework that accounts for biodiversity feedback on crop yields to evaluate the risk and magnitude of intensification traps. Simulations grounded in systematic literature reviews showed that intensification traps emerge in most landscape types, but to a lesser extent in major cereal production systems. Furthermore, small reductions in maximal production (5–10%) could be frequently transmitted into substantial biodiversity gains, resulting in small-loss large-gain trade-offs prevailing across landscape types. However, sensitivity analyses revealed a strong context dependence of trap emergence, inducing substantial uncertainty in the identification of optimal management at the field scale. Hence, we recommend the development of case-specific safety margins for intensification preventing double losses in biodiversity and food security associated with intensification traps.

Rapidly rising global food demand creates a fundamental challenge for agricultural production to meet future needs and ensure food security[1–3]. Past increases in food production have primarily been achieved through cropland expansion and 'conventional intensification' (terms in single quotation marks are defined in Table 1)[4–6]. However, these increases came at the cost of substantial reductions in local biodiversity and associated ecosystem functions[7–9], which can result in a strong negative feedback on yields (that is, productivity) and total agricultural production[10–12].

The negative feedback of biodiversity on yields is especially of importance at high levels of 'management intensities'. First, returns

[1]Department of Computational Landscape Ecology, UFZ—Helmholtz Centre for Environmental Research, Leipzig, Germany. [2]Marine Ecology Department, Lurio University, Nampula, Mozambique. [3]Institute for Resources, Environment and Sustainability, University of British Columbia, Vancouver, British Columbia, Canada. [4]Department of Zoology, University of British Columbia, Vancouver, British Columbia, Canada. [5]Biodiversity Research Centre and IBioS Collaboratory, University of British Columbia, Vancouver, British Columbia, Canada. [6]Environmental Sustainability Research Centre, University of Derby, Derby, UK. [7]Instituto de Investigaciones en Recursos Naturales, Agroecología y Desarrollo Rural, Universidad Nacional de Río Negro, Viedma, Argentina. [8]Instituto de Investigaciones en Recursos Naturales, Agroecología y Desarrollo Rural, Consejo Nacional de Investigaciones Científicas y Técnicas, Viedma, Argentina. [9]Integrated Biosphere Futures, International Institute for Applied Systems Analysis, Laxenburg, Austria. [10]Department of Environmental Studies, University of Colorado Boulder, Boulder, CO, USA. [11]School of Public Policy and Global Affairs, University of British Columbia, Vancouver, British Columbia, Canada. [12]Institute of Geoscience and Geography, Martin-Luther University Halle-Wittenberg, Halle (Saale), Germany. [13]German Centre for Integrative Biodiversity Research (iDiv) Halle-Jena-Leipzig, Leipzig, Germany. ✉e-mail: flinserl@hotmail.com

## Table 1 | Definition of terms

| Term | Definition |
| --- | --- |
| Conventional intensification | Intensification that relies on external inputs (for example, fertilizers and pesticides) within crop monocultures; contrasts with ecological intensification that promotes ecological interactions to increase yield |
| Management intensity | Summary term for (1) the level of conventional intensification and (2) the extent of agricultural land use; both variables are treated as separate dimensions of agricultural land use in our analysis |
| Biodiversity–production relationship | Joint patterns of biodiversity and production arising under different management intensities in a landscape |
| Trade-offs between biodiversity and production | Emerge when increases in management intensity trigger biodiversity loss but still boost production; contrasts with intensification traps |
| Intensification trap | Lose–lose scenario, characterized by biodiversity and production losses resulting from overly high management intensities; characterized by a risk and a maximal production loss (Fig. 2a) |
| Opportunity–cost curve | A curve indicating the maximal biodiversity that can be achieved at a given production level; opportunity costs of increasing production or biodiversity can be derived from the difference of two points on the curve; visualizes trade-offs between biodiversity and production |
| Management–biodiversity–production nexus | The relationships that interlink land management, biodiversity and food production in agricultural landscapes |

per effort decrease at high management intensities because of the saturating response of crop yields to conventional intensification[13]. Likewise, cropland expansion results in diminishing returns as it frequently occurs in marginal areas with lower yield potential owing to unavailability or protection of more suitable sites[5,14]. By contrast, negative impacts of intensification on biodiversity may even increase at high management intensities[15], potentially causing the crossing of tipping points that trigger sudden community breakdowns[16]. The associated loss in crucial services, such as pollination or natural pest suppression, may outweigh direct benefits of higher management intensities and lead to hump-shaped production responses[13,17]. Under such circumstances, high management intensities lead to lose–lose situations instead of the frequently anticipated 'trade-offs between production and biodiversity'[18].

We refer to such lose–lose situations as 'intensification traps' as they are commonly linked to substantial negative legacy effects of past land use[19]. Over-intensification can cause, for example, soil biodiversity and fertility losses that require long-term restoration[20,21] and thereby create barriers that prevent farmers from exiting trap situations. Likewise, above-ground biodiversity shows lagged responses to regenerative agroecological practices and the full recovery of ecosystem functionality requires extensive time periods[22,23]. Furthermore, if yield declines due to loss in biodiversity are misinterpreted as conventional yield gaps (that is, yields being limited by lack of inputs), they can lead to additional intensification and the self-reinforcement of traps. Hence, the avoidance of intensification traps, with their associated losses in biodiversity and food production, and their intransigence to reversal, needs to be a central goal of agricultural management[24].

Despite their importance, the mechanisms driving intensification traps are conceptually not well resolved[13,25]. One associated challenge

is that the crop, soil and biotic characteristics of landscapes are highly variable. This variability is transmitted to 'biodiversity–production relationships' that determine the occurrence of intensification traps. Hence, a crucial step to detect and prevent intensification traps is a clear mechanistic description of how crop, soil and biotic landscape characteristics shape the relationships underpinning the 'management–biodiversity–production nexus'[13].

Our aim in this study was therefore to identify the biophysical mechanisms driving intensification traps and evaluate how the emergence of traps varies with changes of crop, soil and biotic characteristics in agricultural landscapes. Our assessments are based on a novel analytical framework that integrates biodiversity as both predictor and response variable into agricultural planning. A core element of this framework is the conceptualization of five key relationships that together determine the covariation of biodiversity and crop production at the landscape level.

## Five key relationships driving biodiversity and yield

Both crop production and biodiversity, represented here by species richness, depend on the intensity and the spatial extent of agricultural land use[1,10]. These dependencies can be described by five key non-linear relationships influencing crop yield directly or indirectly by mediating biodiversity effects (Fig. 1). These relationships vary in their effect sizes – the change in the response variable across the range of the predictor – and the shape of response curves, that is, the degree of their non-linearity; together, they define biodiversity–production relationships in agricultural landscapes.

(A) Dependency of average yield on the spatial extent of production: The massive expansion of human land use over the last 300 years across most geographic regions has resulted in a scarcity of productive land[5]. Thus, agricultural expansion primarily occurs now in areas with lower yield potential[26,27] and results in a negative, non-linear impact on the average attainable yield in a landscape. The effect size and shape of this relationship thereby depend on the frequency distribution of the yield potential and hence on the heterogeneity of biophysical production conditions (Fig. 1 and Extended Data Fig. 1).

(B) Response of yield to conventional intensification: The saturating response of crop yield to external inputs such as fertilizers or pesticides, which are here primarily considered as conventional intensification, is long established[28]. Both effect size and shape of this relationship depend on biophysical properties and reflect, for example, crop nutrient requirements, nutrient deficiencies in unfertilized soils and the sensitivity of crops to pests.

(C) Dependency of yield on biodiversity: The positive impact of biodiversity on yield is rooted in associated ecosystem functions such as soil nutrient cycling, pollination or natural pest suppression[7,11]. Whereas the effect size of this relationship depends primarily on crop requirements (for example, pollinator dependency[29]), its shape is determined from biological characteristics such as species' effect traits[30] or the degree of functional redundancy. For example, if pest-suppressing predators are primarily generalists and show high redundancy, a concave relationship can be expected. By contrast, the requirement of many specialized predators for effective pest suppression can be expected to result in convex relationships[31].

(D) Biodiversity responses to conventional intensification: Natural communities are known to respond negatively to eutrophication and pesticide applications[32,33]. The effect size of conventional intensification on biodiversity depends on the sensitivity of natural communities as well as on the type of input (for example, pesticides and fertilizers), which is often related to crop type. The shape of this relationship depends on the sensitivity of natural communities[34], as high proportions of sensitive species

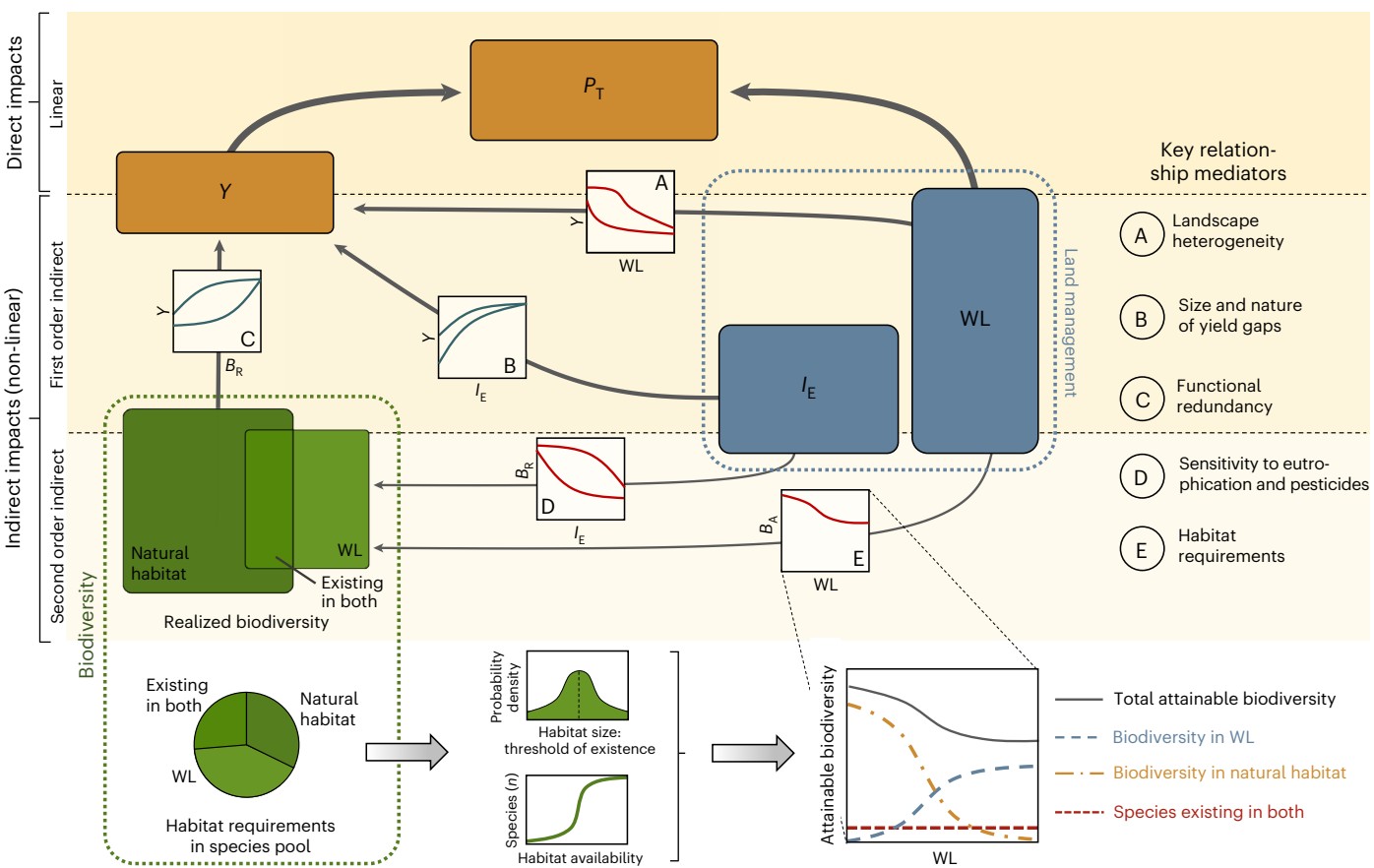

**Fig. 1 | Conceptual overview of five key relationships mediating the impact of land management (blue boxes) on biodiversity (green) and agricultural production (brown).** Land management is characterized by the level of conventional intensification effort ($I_E$) and the proportion of land used for agriculture (that is, WL). Crop yield, that is, production per area ($Y$), depends on land-management features directly (plots A and B) and indirectly (C–E) via biodiversity. The effect size (change in the response variable across the range of the predictor) and the shape of these five relationships will vary across landscapes with crop, soil and biotic characteristics and key relationship drivers (listed in the top right). In our framework, the response of the attainable biodiversity ($B_A$) to changes in land use results from habitat requirements of species in the regional species pool (bottom left) and their required minimum habitat size (bottom centre). Realized biodiversity ($B_R$) is estimated by subtracting the negative impact of conventional intensification from $B_A$.

result in rapid and hence convex responses whereas high proportions of tolerant species trigger a concave shape (Fig. 1).

(E) Biodiversity responses to changes in the extent of agricultural land use: Biodiversity responses to changes in land-use types emerge from the responses of individual species that reside in a landscape or may colonize it from the regional species pool[32,35]. A defining parameter for their presence is the minimum amount of suitable habitat that is required by a species for persistence[36]. We, therefore, categorized species based on their ability to colonize agricultural land as well as natural and semi-natural habitats (further summarized as natural habitat). In addition, each species exhibits a threshold for the proportion of suitable habitat that is required for its persistence (Fig. 1, bottom). Together, these traits characterize species' responses and thereby also the maximal species richness that can be reached under different land uses.

Hence, the effect sizes and shapes of these five key relationships reflect the crop, soil and biotic characteristics of a landscape and define the responses of biodiversity and crop production to increasing management intensities. We evaluated how changes in these relationships impact the emergence of intensification traps, using a biodiversity–production model and a set of systematic literature reviews. Literature reviews were implemented for each model constant with the aim of capturing their natural variability across agricultural landscapes. Reviews included a restricted meta-analysis (see Section A2 of

Supplementary Information for details) that was complemented by a snowball search to balance among crop types and geographic regions, resulting per model constant in over ten datasets for parametrization. This allowed us to (1) evaluate the occurrence of intensification traps across artificial landscapes, which were created stochastically to reflect the natural variability of our five key relationships; (2) establish three archetypal case studies contextualizing our results; and (3) implement a systematic sensitivity analysis to explore the model's parameter space and identify mechanistic drivers of intensification traps at the landscape scale.

## Results and discussion

### Biodiversity–production patterns in agricultural landscapes

Our analyses evaluated the risk of entering intensification traps as well as their associated maximal production loss (Fig. 2a). We found that the implementation of the highest management intensities resulted in intensification traps in 73% of artificial landscapes. This trap prevalence directly emerges from the parametrization of our five key relationships based on the variability of real-world data recorded in our literature reviews. Hence, artificial landscapes represent the range of potential crop, soil and biotic characteristics, without being proportionate to the current prevalence of global production systems. Both risk and maximal production loss associated with traps are strongly driven by the effect size of biodiversity on agricultural yields in a given landscape (Fig. 2c,d). Yet, both these relationships show a high degree of scattering. This variability results from the joint impact of multiple drivers and

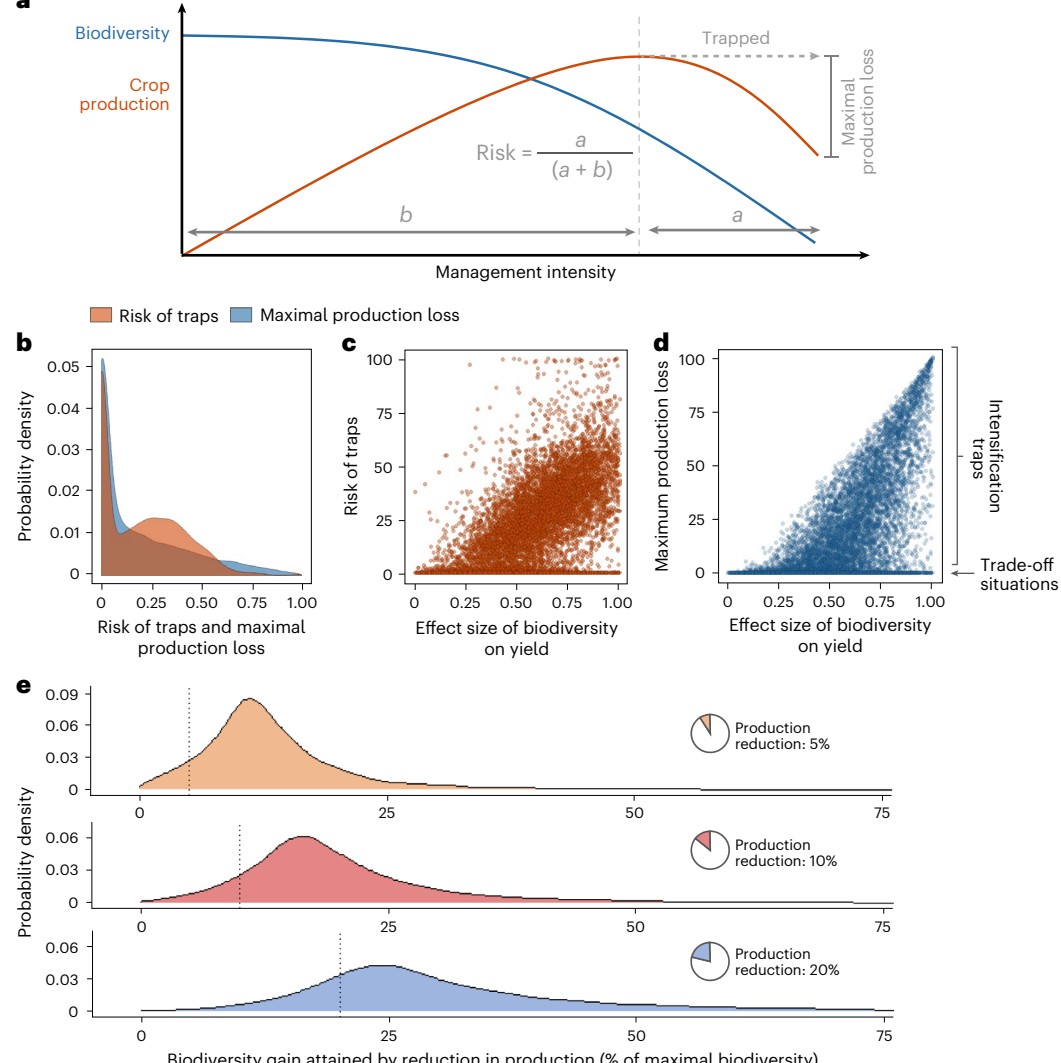

**Fig. 2 | Characterization of intensification traps in agricultural landscapes.**
**a**, Onset of intensification traps, definition of their risk and their associated maximal production loss. Both risk and maximal production loss are scaled from 0 to 1, with 1 denoting the highest theoretically possible value. **b**, The distribution of the risk of intensification traps and associated maximal production losses across 10,000 stochastically generated artificial landscapes. Artificial landscapes reflect the variability of the five key relationships (Fig. 1) as recorded in a literature review for each of the model parameters. Parameter values have been range transformed. **c**,**d**, The risk of intensification traps (**c**) and associated

maximum yield losses (**d**) in 10,000 artificial landscapes are dependent on the effect size of biodiversity on yields. Each point represents a landscape. Note that landscapes with high risk (>80% of possible land uses) were occurring in less than 1% of all cases and were driven by situations when biodiversity peaked at intermediate to high contribution of working lands. **e**, The biodiversity gains attained by decreasing maximum production by 5%, 10% and 20%. Values to the right of the dashed lines indicate greater biodiversity gains than production losses (small-loss large-gain situation).

also leads to situations in which trade-offs rather than intensification traps prevail (Fig. 2d).

Three archetypal landscapes were chosen to provide contrasting examples of production systems that are of importance for global food security and largely differ in their five key relationships underlying intensification responses. These archetypal case studies also showed a large variation in the occurrence of intensification traps (Fig. 3). In two of the three landscapes, the US wheat belt and the Southeast Asian rice scenarios, even the highest management intensities did not trigger intensification traps despite the presence of positive biodiversity effects. By contrast, in the sub-Saharan small-holder scenario, a system with higher crop diversity and pollinator dependence, crop production was substantially reduced by high management intensities (Fig. 3). These contrasting responses emerge directly from differences in the parametrization of the five key relationships defined in our framework (see Extended Data Fig. 1 and below for an explanation of mechanisms). This suggests that globally dominant cereal production systems are less

sensitive to biodiversity loss and intensification traps (crop type was much more important for parametrization than region; see Section A2 of Supplementary Information). However, the higher sensitivity of systems with a greater risk of intensification traps does not imply that the crop, soil and biotic characteristics of these landscapes are less 'favourable'. Instead, a higher sensitivity signifies that these systems require a careful integration of biodiversity into management schemes to avoid biodiversity-driven yield gaps.

Our analytical framework also allowed us to establish an 'opportunity–cost curve' of biodiversity and production for each individual landscape (Fig. 3). These curves depict the maximum biodiversity that can be attained at a certain production level. We found a strong prevalence of non-linear shapes for opportunity–cost curves across our analyses. Hence, small reductions in production can be 'traded off' for large biodiversity gains, which is exemplified by the rice case study, in which reducing the maximal crop production by 5% results in a doubling of biodiversity. Such small-loss large-win trade-offs also occur in

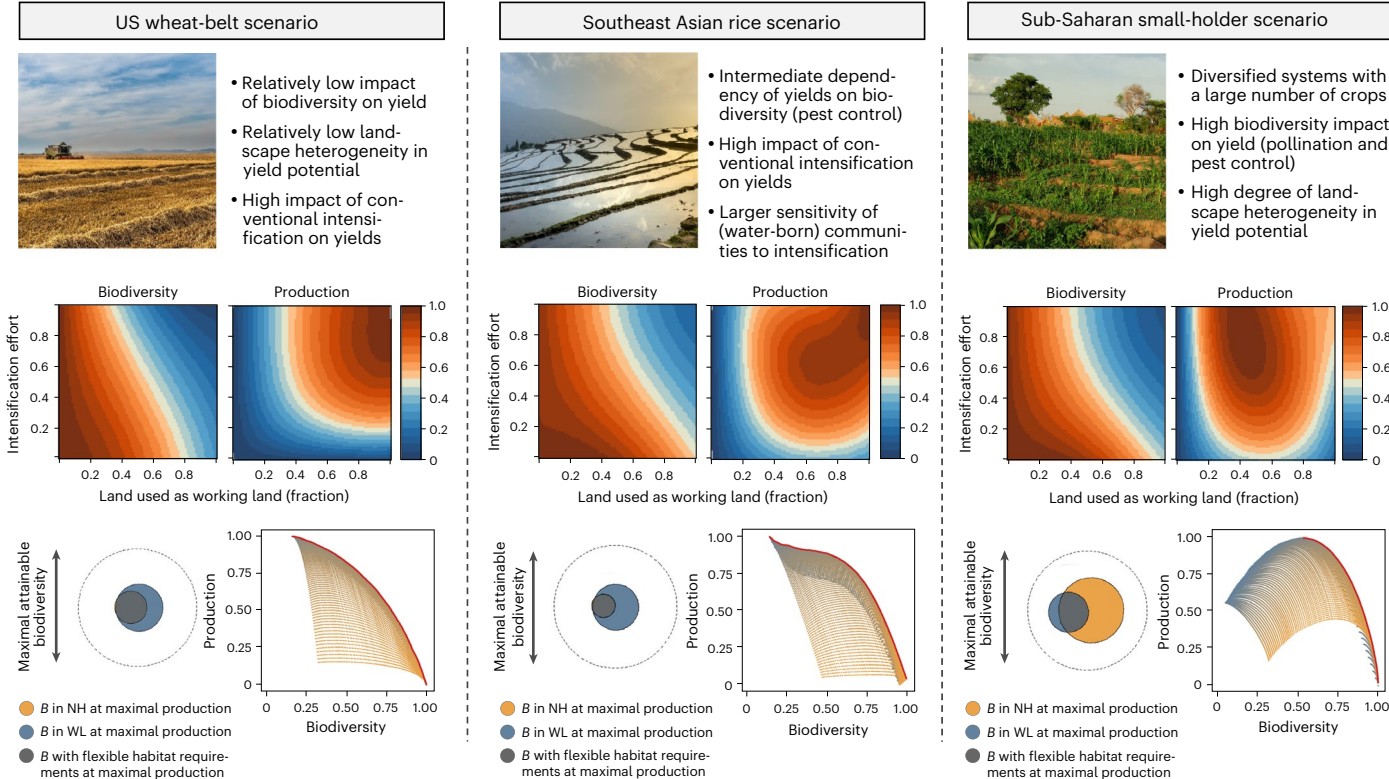

Fig. 3 | Analysis of three selected archetypal landscapes. Exemplary case studies were chosen as they represent production systems of large importance for global food security that vary in their reliance of yield on biodiversity. In the middle row, responses of biodiversity (left) and agricultural production (right) to changes in management intensity (that is, conventional intensification and the proportion of agricultural land-use) are presented. In the bottom left, maximal attainable biodiversity (dotted circle) is depicted compared with the biodiversity maintained under the land management that leads to the highest agricultural production. Opportunity–cost curves (bottom right), which show the highest attainable biodiversity for each production level, are represented by red lines whereas model scenarios are shown as points coloured based on their conventional intensification effort (high, grey; low, yellow). B, biodiversity; P, production; NH, natural habitat. All axis units are range transformed. Credits, top row: left, josealbafotos/Pixabay; centre, Quangpraha/Pixabay; right, TG23/iStock.

the other case studies (Fig. 3) and are predominant across our artificial landscapes. Hence, reductions by 5–10% of maximal total production result frequently in disproportionately larger biodiversity gains, even in landscapes where intensification traps do not occur (Fig. 2e).

## Understanding the drivers of intensification traps

The goal of our sensitivity analysis was to identify landscape characteristics that increase the likelihood of trap emergence. Systematic changes in the parameters defining the five key relationships of our conceptual framework revealed that both effect sizes and relationship shapes had a large impact on the risk of intensification traps (Fig. 4). These results show that intensification traps emerge in situations in which indirect consequences of biodiversity loss on yields outweigh the direct production-stimulating effects of increasing management intensities. Hence, the impacts of changing effect sizes and relationship shapes can be explained by their moderation of (1) the yield penalties caused by biodiversity loss and (2) the direct production benefits of increasing management intensities.

Direct production benefits of increased management intensities are governed by relationships A (yield responses to expansions of agriculture) and B (yield responses to conventional intensification). Relationship A is shaped by the distribution of the yield potential within a landscape, which is defined by its average and standard deviation. Both decreasing the average and increasing the standard deviation of the yield potential lead to a larger area of land that can attain only low yields (Extended Data Fig. 2). Consequently, maintaining or restoring marginal areas as natural habitats is linked to relatively small direct negative yield effects[37], which can more easily be compensated by biodiversity-mediated yield benefits.

Notably, this conclusion is based on the assumption that the potential of an area to support crop yields and biodiversity are uncorrelated as their covariation can affect land-use trade-offs[38]. In the case of intensification–yield relationships, represented by relationship B, reductions in effect size and increases in non-linearity have similar consequences (Fig. 4). Both lower direct production benefits attained from increasing management intensities from high to very high levels and therefore increase the risk of intensification traps.

Production gains resulting from increased biodiversity are determined by relationship C. Naturally, a higher effect size of this relationship increases biodiversity-mediated yield benefits and hence also the risk of intensification traps under conventional intensification (Fig. 4). Furthermore, strongly convex curves require that high levels of biodiversity are maintained to support agricultural production effectively. Such high levels of biodiversity can be attained only by substantial reductions in management intensities and pay off only when the effect size of biodiversity on yields is high (Fig. 4). Conversely, strongly concave biodiversity–yield relationships allow yields to benefit even from relatively low levels of biodiversity. Therefore, strongly concave relationships increase the risk of intensification traps but at the same time lower the biodiversity levels that are required to overcome trap situations (Fig. 4). Field and experimental studies show that the shape of biodiversity–ecosystem function relationships can vary greatly from strongly concave[39] to convex responses[30]. However, the associated far-reaching impacts on the management of natural habitats in agricultural landscapes are rarely considered.

Relationship D, the response of biodiversity to conventional intensification, varies in agricultural landscapes[40], for example, with the

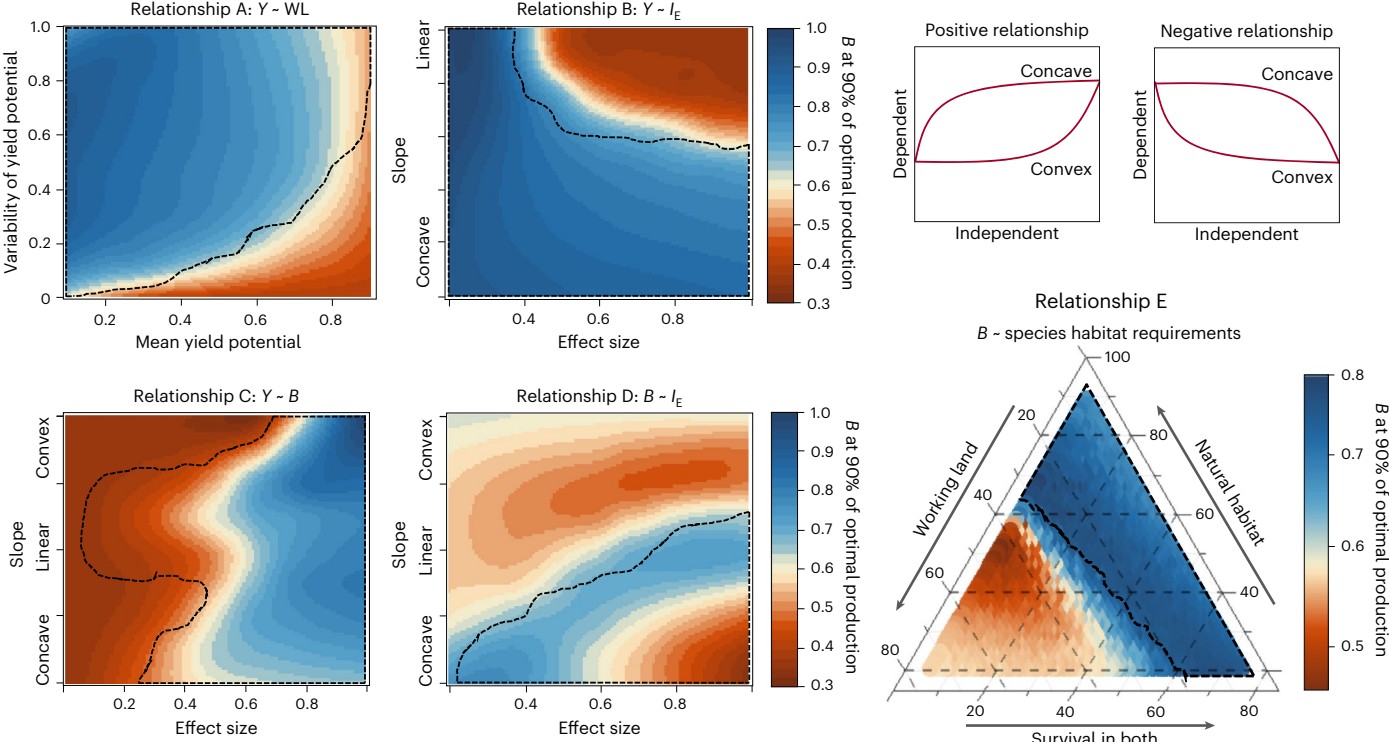

**Fig. 4 | Systematic sensitivity analysis of how changes in the five key relationships presented in Fig. 1 affect biodiversity–production relationships.** Biodiversity remaining in an agricultural landscape when at least 90% of maximum attainable production is achieved. In five sensitivity analyses (relationships A–E), the model constants describing one of the five key relationships were systematically modified over a predefined range (Supplementary Table 2). The dotted area represents conditions in which intensification traps emerge at maximal management intensity (that is, at maximal conventional intensification and agricultural land expansion). Yield potential in A refers to the highest attainable yield in a given area and was standardized from 0 to 1 in each landscape. Shapes of positive relationships apply to panels B–C, and those of negative relationships, to panel D (top right).

sensitivity of natural communities to pesticides and eutrophication. We found that a shift from concave to convex negative relationships, representing a higher community resistance, generally lowered the risk of intensification traps (Fig. 4). Convex relationships require that conventional intensification is drastically reduced before biodiversity can recover. Such large reductions in inputs are linked to a strong reduction in direct benefits of intensification, and the likelihood that they result in a net production increase is rather low. By contrast, the effect size of this relationship increases the risk of intensification traps. This pattern emerges because high effect sizes raise the amount of biodiversity that is lost by intensification, leading to stronger negative feedback on yields (Extended Data Fig. 3).

Finally, we considered changes in the regional species pool and its impact on biodiversity–production trade-offs (Fig. 4). The characteristics of the regional species pool are pivotal for biodiversity–ecosystem function relationships and local biodiversity patterns[35,41] but receive little consideration in landscape planning. We found that changes in habitat preferences in the regional species pool trigger a sudden switch in the land-use strategy that optimizes crop production (Fig. 4 and Extended Data Fig. 4). If many species rely on natural habitats, the integration of these patches into landscapes substantially increased biodiversity and its associated benefits for yields. However, if many species required only working lands as habitat, yield benefits associated with natural habitats decreased. Furthermore, direct effects of conventional intensification on species in fields are much stronger than spillover effects on species in adjacent habitats[42], which is integrated into our land-use model (Methods). Consequently, if a large proportion of species live only in working lands, a lower level of intensification is required to maintain these species, making the reconciliation of biodiversity and crop production more challenging.

## From models to practice

Our analytical framework allows evaluating mechanistic biophysical drivers of intensification traps, which represent a substantial challenge for global food production and security[7,13,17]. However, agricultural systems show a high level of inherent complexity and a general assessment of intensification traps requires a number of simplifications. For example, both conventional intensification and biodiversity are in their essence multidimensional[9,12,43] but their realistic representation would result in much higher model complexity, hampering conceptual advancements (see Section A3 of Supplementary Information for a detailed discussion). Hence, underlying model assumptions need to be taken into consideration when extrapolating our results to farm, landscape and regional scales.

At the farm level, financial cost–benefit relationships are key determinants of individual decision-making processes[44]. In this context, a clear understanding of intensification traps is crucial to avoid lose–lose situations that reduce production and biodiversity while raising farmers' spending on pesticides and fertilizers. However, a precise determination of trap onsets requires detailed process-based information, which realistically cannot be compiled for each individual farm. The resulting uncertainty farmers are facing makes it economically advisable to follow precautionary principles and maintain management intensities slightly below anticipated optima. This would require, due to non-linear opportunity–cost curves, only small decreases in maximum farm revenue but substantially reduce the risk of intensification traps and long-term losses in fertility[45]. Hence, such safety margins restricting intensification would help to prevent financially highly detrimental lose–lose scenarios and enhance local biodiversity as a positive side effect.

In addition, farmers, in reality, may choose among a large variety of different land management practices, which our model framework simplifies into two dimensions (that is, land expansion and conventional intensification). An important alternative approach is ecological

intensification[46], which is frequently associated with practices such as intercropping or planting of cover crops[17,47]. These approaches are typically linked to higher labour but lower input costs[48]. Therefore, a thorough comparison of conventional and ecological intensification, which is beyond the scope of this study, would need to incorporate labour and other costs as an additional management dimension to identify optimal solutions in cost–benefit analyses[13].

At the landscape level, the spatial arrangement of natural habitats represented by landscape structure is an essential factor influencing biodiversity–production relationships[36]. Landscape structure directly regulates the impact of natural habitats on agricultural yields[49] and affects, by defining connectivity, the amount of realized biodiversity in local habitat patches as well as the size of regional species pools.[35,41,50]. We considered in our framework only the proportion of different land uses in a landscape and their impact on habitat availability for species in the regional special pool. However, specific landscape structure as well as other drivers of the size and trait frequencies in regional species pools (for example, evolutionary history linked to past agricultural practices[51]) can strongly moderate biodiversity–yield relationships and should be considered in the management of agricultural landscapes.

At the regional level, policies are required to prevent the biodiversity-dependent yield decreases shown in this study. In this context, the bimodal pattern in the distribution of the risk of intensification traps (Fig. 2b) and optimal management intensities (Extended Data Figs. 5 and 6) require further attention. These bimodal patterns result from the occurrence of two local optima of total production in the management-opportunity space of many agricultural landscapes. One of these optima is driven by the positive effects of conventional intensification on yield while the other emerges from yield benefits linked to high levels of biodiversity. Management intensities located between these optima have lower yields because the reductions in management intensities are insufficient to facilitate a functionally meaningful biodiversity recovery. Hence, policies that enforce or result in weak ecological minimum requirements (for example, a 3% non-productive farm area as a requirement for agricultural subsidies in the Common Agricultural Policy 2023–2027[52]) might paradoxically risk promoting low points between production optima in many landscapes.

## Conclusions

The prevention of intensification traps, which are characterized by a double loss of biodiversity and agricultural production, will be a crucial task for the sustainable management of life on earth[53]. We evaluated here intensification traps triggered by biodiversity loss, but our analyses can easily be expanded to other drivers such as soil degradation or salination. Our results highlight that the risk of intensification traps is increased by (1) a larger effect size of biodiversity on yields, (2) a stronger reliance of beneficial species on natural habitats and (3) stronger and more immediate responses of natural communities to conventional intensification. Furthermore, the risk of traps is decreased by stronger and more linear impacts of intensification on yields and with higher averages and less variable distributions of yield potentials in agricultural landscapes. Due to this complexity, it is difficult to quantify optimal management intensities at the farm level, and advisable to follow precautionary principles to avoid lose–lose scenarios. Furthermore, we found that across the vast majority of agricultural landscapes, small reductions in agricultural production can be translated into disproportionally larger biodiversity gains. These small-loss large-gain scenarios offer attractive opportunities to increase biodiversity in agricultural landscapes and can, along with a careful consideration of conservation targets, help to reconcile seemingly conflicting land-use targets.

## Methods

### Model framework

In our framework designed for landscape-scale assessments[54], land management comprises two key aspects, (2) the proportion of land used for agricultural production (that is, working land (WL)) and (2) the level of conventional agricultural intensification ($I_E$). $I_E$ represents the external inputs associated with conventional intensification such as fertilizer and pesticide use. Conventional intensification contrasts with ecological intensification, which includes the establishment of semi-natural habitat patches as one out of a large array of agroecological practices[46]. Our framework accounted for the possibility of regulating the proportion of semi-natural habitat patches, defined as $1 - WL$. However, we decided to exclude other agroecological practices as those are associated with substantially higher labour inputs[48] and the integration of labour costs was beyond the scope of this study (see Sections A2 and A3 of Supplementary Information).

### Total crop production

Total agricultural production ($P_T$) is computed as

$$P_T = WL\, Y \tag{1}$$

where $Y$ denotes the yield, that is, production per area. The direct dependency of yields on land management (relationships A and B) and the total biodiversity in a landscape ($B_T$, relationship C) as well as the indirect dependency of yields on the response of biodiversity to management (relationships D and E) is described by

$$Y = Y_{Max} f_{I_E}^Y (I_E) f_{B_T}^Y (B_T) f_{WL}^Y (WL) \tag{2}$$

$$B_T = f_{I_E\,WL}^B (I_E, WL) \tag{3}$$

where $Y_{Max}$ is the maximal attainable total production, while $f_{I_E}^Y (\cdot), f_{B_T}^Y (\cdot), f_{WL}^Y (\cdot)$ and $f_{I_E\,WL}^B (\cdot)$ are four functions with five predictor terms that represent the five key relationships defined in Fig. 1.

The three terms that define the impact of $I_E$ and $B_T$ are each described by two model constants, the effect size of the independent variable on the dependent variable and the shape of the relationship (convex, concave). Effect size is confined to values between 0 and 1 and indicates the proportional change in the response variable if the predictor increases from 0 to 1 (that is, its range). The relationship slope is scaled from −1 to 1, where 0 represents a linear, 1 a convex and −1 a concave slope (see Section A1 of Supplementary Information and Supplementary Fig. 1).

Spatial elements are integrated into the functions in equations (2) and (3) that involved WL as a predictor. The impact of WL on yield (Fig. 1a) is based on the assumption that areas with a higher yield potential are first used for crop production and derived from the mean and the variance of the yield potential within a landscape (see Section A1 of Supplementary Information). Moreover, the response of biodiversity to changes in WL is based on the regional species pool (Fig. 1, bottom), which is recognized as a key determinant of biodiversity–ecosystem function relationships[35,41]. Each species in the species pool was linked to habitat requirements, which determine individual species' responses to changes in agricultural land use (for details, see Section A1 of Supplementary Information).

We computed for each analysed landscape the biodiversity and production attained under different land management options. A land management option represented a combination of WL and $I_E$, and for each landscape, 10,201 land management options were analysed (all possible combinations with WL and $I_E$ ranging from 0 to 1 in steps of 0.01). For simplicity, we range transformed attained biodiversity and total production across all 10,201 simulated land management scenarios to scale outputs from 0 to 1 within each landscape.

### Implemented analyses

Our assessments of biodiversity–production relationships were based on three distinct approaches including (1) a stochastic landscape generation procedure, (2) the evaluation of archetypal case studies and (3) a systematic sensitivity analysis.

The variability of biodiversity–production relationships across landscapes was assessed by generating 10,000 artificial landscapes in a bootstrapping procedure. The analysis was based on a set of literature reviews to identify means and standard deviations of all model constants parameterizing equations (2) and (3). We accumulated, for the parametrization of individual model constants, between 11 and 26 datasets (Supplementary Table 2) and then used means and standard deviations to create a normal distribution for each model constant. An artificial landscape was created by randomly drawing a value for each model constant from its respective distribution, and analysed by establishing biodiversity and production outputs for each of the 10,201 land-management scenarios.

Archetypal case studies included a US wheat-belt scenario, a Southeast Asian rice scenario and an African small-holder scenario, which were based on intercropping and more diversified crop cultivation. They were used for contextualization of model results, and their parametrization is described in detail in Section A2 of Supplementary Information.

The response of model outputs to changes in individual model constants was investigated in a systematic sensitivity analysis. For each model constant, 100 landscapes were simulated and the target model constant was gradually changed from an upper to a lower range while all other model constants were maintained at mean literature values. Extreme but realistic values were chosen for the ranges of model constants (Supplementary Table 2). Biodiversity–production responses to different land-management scenarios were established for each landscape to identify the most important mechanisms driving intensification traps. All analyses were developed and implemented in R version 4.1.0 (ref. 55).

### Reporting summary

Further information on research design is available in the Nature Portfolio Reporting Summary linked to this article.

## Data availability

No original data were used for this work.

## Code availability

Annotated versions of the model scripts, input data and a help document providing general instructions for the implementation of the model are provided under https://github.com/alfredburian/Intensification-traps.

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

## Acknowledgements
A workshop initiating this article was funded by the University of British Columbia (Grants for Catalyzing Research Clusters to C.K. and N.R. through the Biodiversity Research Centre).

## Author contributions
A.B., C.K., L.A.G., Z.M., N.M., M. Beckmann and R.S. developed the conceptual foundation of the study; A.B., M. Bulling and T.K. established the model code; A.B. and J.S.-T.W. implemented the meta-analyses; and all authors contributed to the establishment of the paper.

## Funding

## Competing interests
The authors declare no competing interests.

## Additional information
**Extended data** is available for this paper at https://doi.org/10.1038/s41559-024-02349-0.

**Correspondence and requests for materials** should be addressed to Alfred Burian.

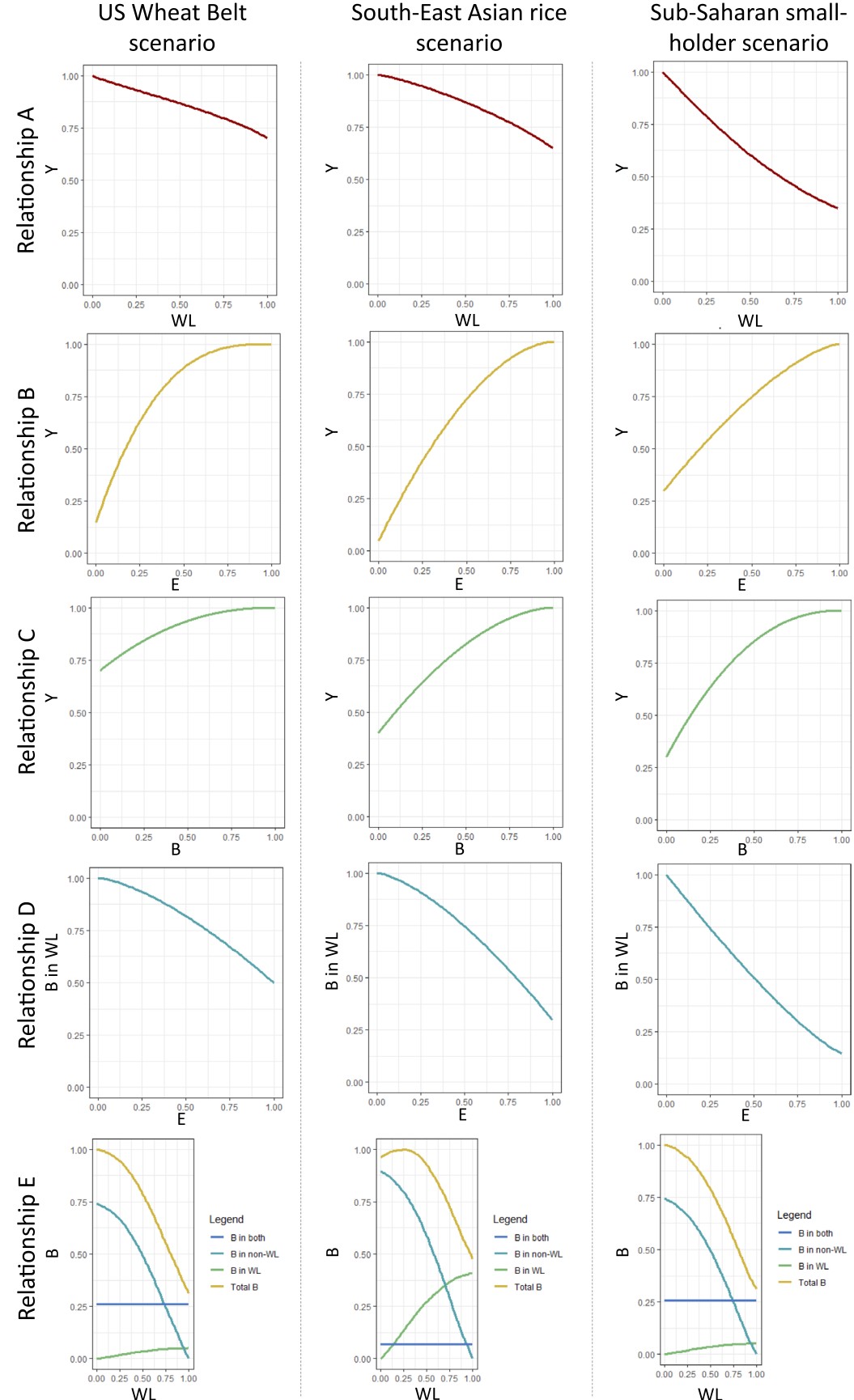

**Extended Data Fig. 1 | Visualisation of the five key relationships for the three archetypal case-studies.** B stands for biodiversity, WL for working land (that is proportion of landscape used for crop production), Y for yield, non-WL for semi-natural and natural habitat (that is non-working land).

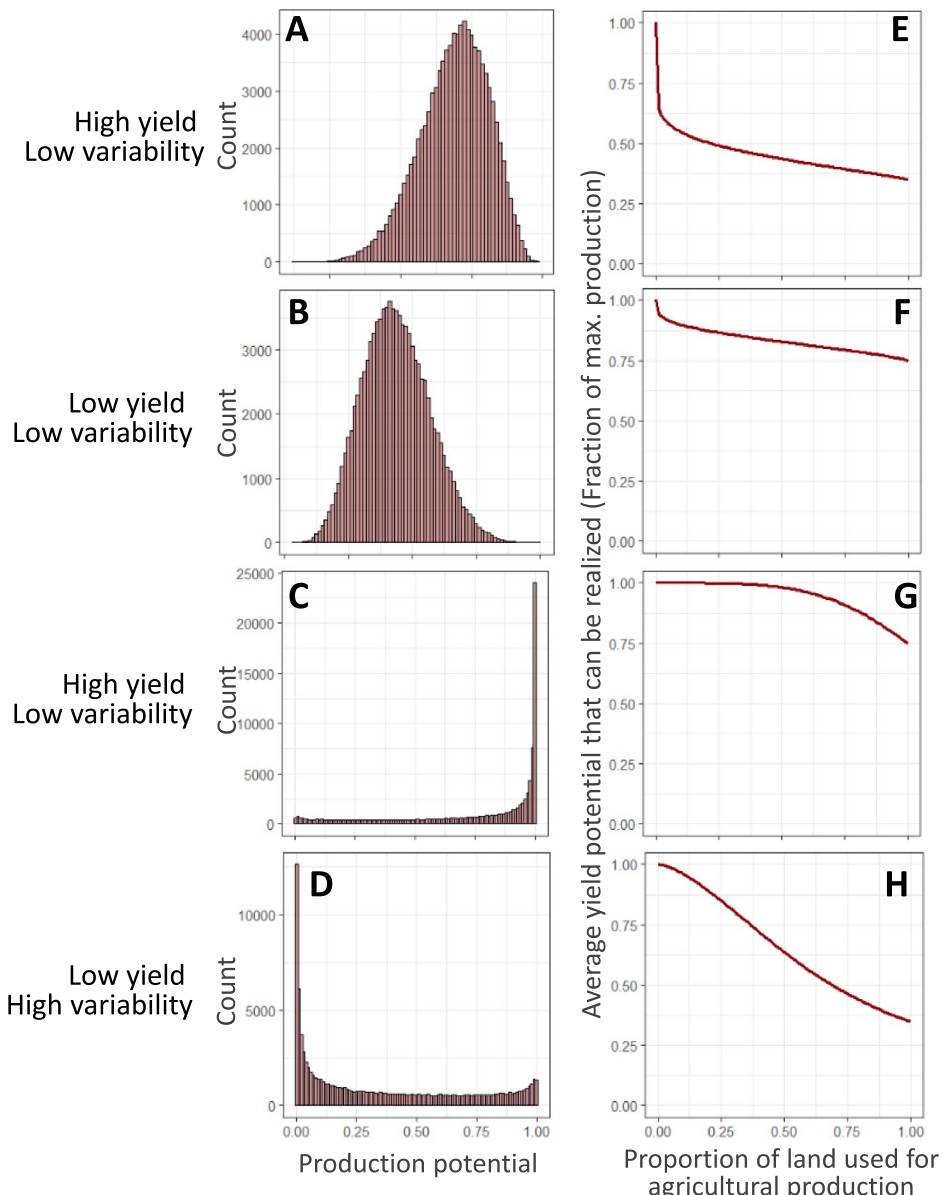

**Extended Data Fig. 2 | Examples for the distribution of production potential in agricultural landscapes. A**–**D** display distributions and **E**–**H** the respective relationships between average potential yield of used fields and proportion of land used for agriculture (curve is derived from distribution). Parameter settings for mean yield and variance of yields (ranges from 0 to 1) were respectively 0.75 and 0.05 (high yield, low variability), 0.25 and 0.05 (low yield, low variability), 0.75 and 0.5 (high yield, high variability) and 0.25 and 0.5 (low yield, high variability).

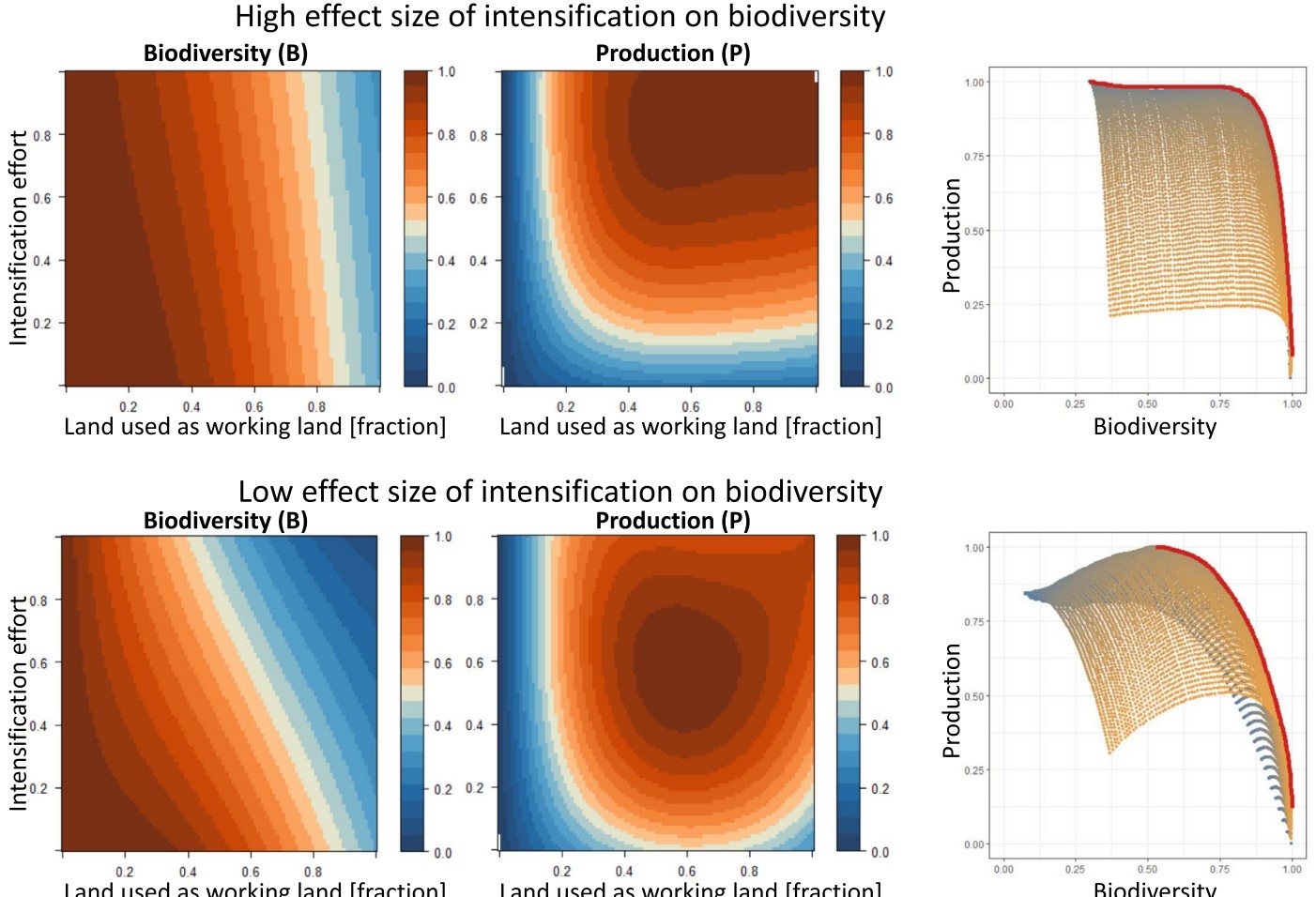

**Extended Data Fig. 3 | The dependency of biodiversity and total production on the intensity of conventional intensification and the extent of working lands in two artificial landscapes.** The two artificial landscapes have been parametrised based on mean literature values for four of the five key relationships and only differ in the parametrisation for relationship D, defining biodiversity responses to conventional intensification. Both landscapes show a linear impact of intensification on biodiversity but landscape 1 (top) was characterised by an effect size of 0.2 whereas the effect size in landscape 2 (bottom) was 0.8. The consequence is that in landscape 1, intensification traps cannot emerge because the negative impact of yield on biodiversity is to small that negative biodiversity feedback-effects on yields overcome the positive effect of intensification on yields. A contrasting situation is found in landscape 2.

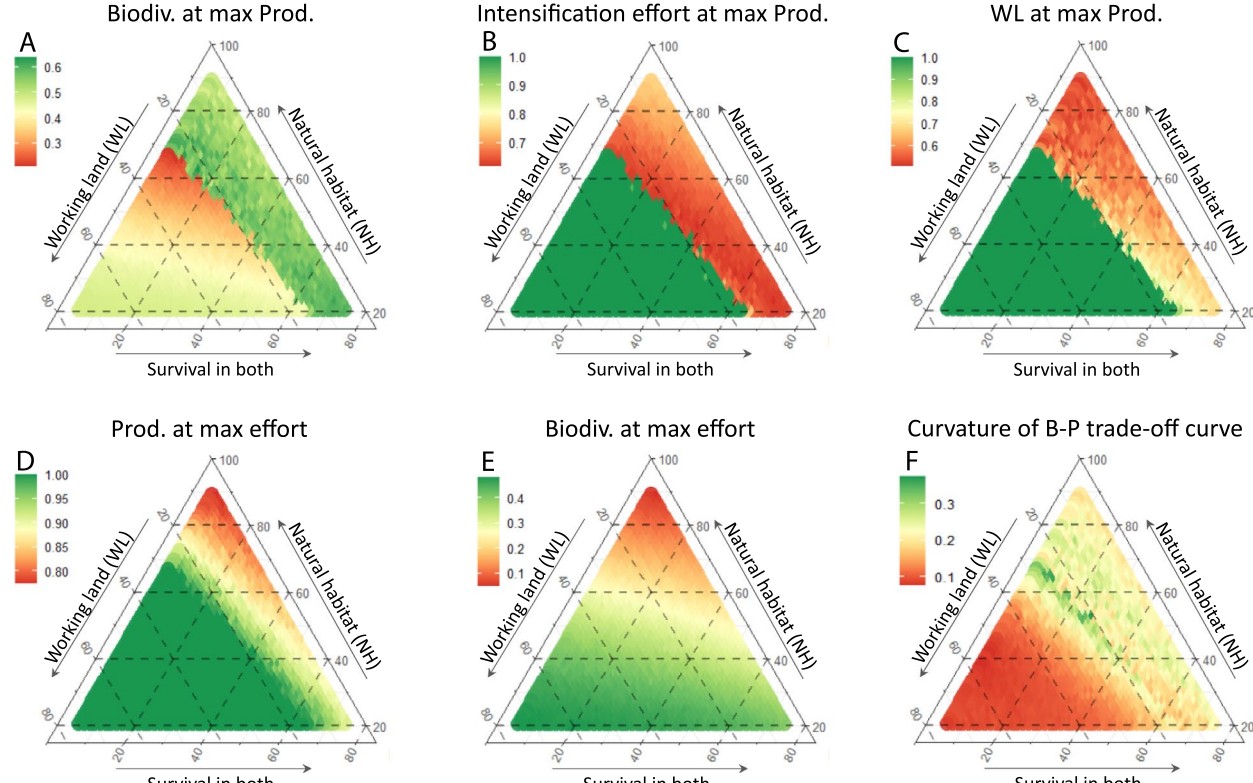

**Extended Data Fig. 4 | Model responses to changes in habitat requirements of species in the regional species pool.** Habitat requirements are defined as inhabiting (i) natural habitat, (ii) working lands or (iii) the ability to survive in both. Displayed as response variables are (**a**) biodiversity in the scenario with the highest total crop production, (**b**) the conventional intensification effort in the scenario with the highest production, (**c**) the proportion of land used as working land in the scenario with the highest production, (**d**) the production achieved in the scenario with the maximum management effort (that is conventional intensification and agricultural land use are both at their maximum), (**e**) the biodiversity maintained in the scenario with the maximum management effort and (**f**) the curvature (that is measure for exponential nature) of biodiversity-production trade-off curves.

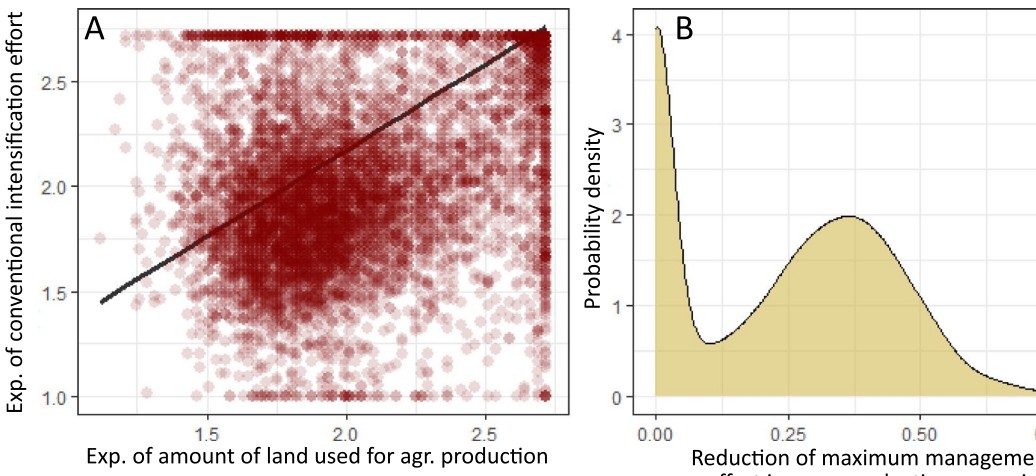

**Extended Data Fig. 5 | The land management options that support maximum total food production in 10000 artificial landscapes.** (**a**) The relationship between conventional intensification and the amount of land used for agricultural production, which represent the two components of land management considered in our study. Plotted is the land management that leads to maximal food production. Each point represents an individual landscape. The two land management components were positively related. The result of a type 2 regression ($R^2 = 0.36$, $p < 0.001$) is depicted as black line. (**b**) The distribution of how strongly optimal land management (highest production) deviates from the maximum management scenario (maximal conventional intensification and agricultural land-use). The difference has been calculated as Euclidian distance and is scaled from 0 to 1 with the latter representing the largest possible distance.

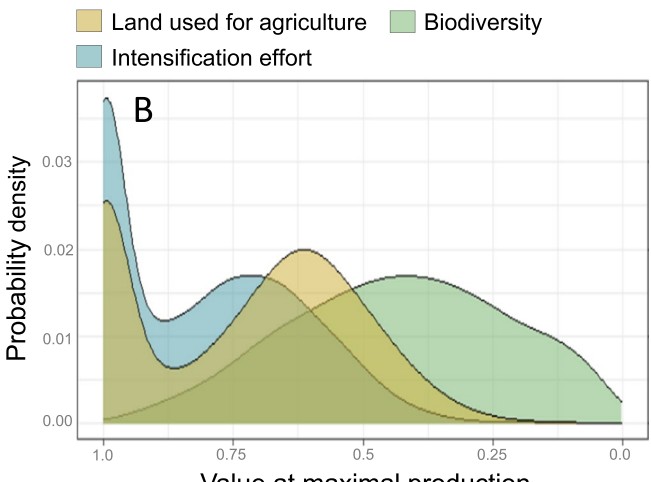

**Extended Data Fig. 6 | Variability of conditions supporting maximal agricultural production.** Displayed are the distributions of biodiversity values, the amount of land used for agricultural production and the conventional intensification effort that support maximum total food production in 10000 artificial landscapes. Biodiversity values are scaled to the maximum biodiversity that can be reached in a landscape.

# Reporting Summary

## Statistics

For all statistical analyses, confirm that the following items are present in the figure legend, table legend, main text, or Methods section.

| n/a | Confirmed | |
|---|---|---|
| ☒ | ☐ | The exact sample size (*n*) for each experimental group/condition, given as a discrete number and unit of measurement |
| ☒ | ☐ | A statement on whether measurements were taken from distinct samples or whether the same sample was measured repeatedly |
| ☒ | ☐ | The statistical test(s) used AND whether they are one- or two-sided *Only common tests should be described solely by name; describe more complex techniques in the Methods section.* |
| ☒ | ☐ | A description of all covariates tested |
| ☐ | ☒ | A description of any assumptions or corrections, such as tests of normality and adjustment for multiple comparisons |
| ☐ | ☒ | A full description of the statistical parameters including central tendency (e.g. means) or other basic estimates (e.g. regression coefficient) AND variation (e.g. standard deviation) or associated estimates of uncertainty (e.g. confidence intervals) |
| ☐ | ☒ | For null hypothesis testing, the test statistic (e.g. $F$, $t$, $r$) with confidence intervals, effect sizes, degrees of freedom and $P$ value noted *Give P values as exact values whenever suitable.* |
| ☒ | ☐ | For Bayesian analysis, information on the choice of priors and Markov chain Monte Carlo settings |
| ☒ | ☐ | For hierarchical and complex designs, identification of the appropriate level for tests and full reporting of outcomes |
| ☒ | ☐ | Estimates of effect sizes (e.g. Cohen's *d*, Pearson's *r*), indicating how they were calculated |

*Our web collection on statistics for biologists contains articles on many of the points above.*

## Software and code

Policy information about availability of computer code

| | |
|---|---|
| Data collection | No software was used for data collection |
| Data analysis | All models that were constructed for this manuscript were implemented in R, version 4.1.0. No other software was used. All R scripts are downloadable under https://github.com/alfredburian/Intensification-traps. |

For manuscripts utilizing custom algorithms or software that are central to the research but not yet described in published literature, software must be made available to editors and reviewers. We strongly encourage code deposition in a community repository (e.g. GitHub). See the Nature Portfolio guidelines for submitting code & software for further information.

## Data

Policy information about availability of data

All manuscripts must include a data availability statement. This statement should provide the following information, where applicable:
- Accession codes, unique identifiers, or web links for publicly available datasets
- A description of any restrictions on data availability
- For clinical datasets or third party data, please ensure that the statement adheres to our policy

No original data was used for this work.

## Research involving human participants, their data, or biological material

Policy information about studies with human participants or human data. See also policy information about sex, gender (identity/presentation), and sexual orientation and race, ethnicity and racism.

| | |
|---|---|
| Reporting on sex and gender | NA |
| Reporting on race, ethnicity, or other socially relevant groupings | NA |
| Population characteristics | NA |
| Recruitment | NA |
| Ethics oversight | NA |

Note that full information on the approval of the study protocol must also be provided in the manuscript.

# Field-specific reporting

Please select the one below that is the best fit for your research. If you are not sure, read the appropriate sections before making your selection.

☒ Life sciences ☐ Behavioural & social sciences ☐ Ecological, evolutionary & environmental sciences

For a reference copy of the document with all sections, see nature.com/documents/nr-reporting-summary-flat.pdf

# Life sciences study design

All studies must disclose on these points even when the disclosure is negative.

| | |
|---|---|
| Sample size | Our simulations were based on stochastic landscape creation process. This process was based on the parametrisation of model constants with at least 10 independent data derived from literature per model constant. |
| Data exclusions | No data was excluded from the analyses. |
| Replication | Simulations of stochastically created landscapes were repeated 10,000 times. |
| Randomization | As our analytical results were based on a modelling approach, no randomisation procedures were necessary. |
| Blinding | As our analytical results were based on a modelling approach, no blinding procedures were necessary. |

# Reporting for specific materials, systems and methods

We require information from authors about some types of materials, experimental systems and methods used in many studies. Here, indicate whether each material, system or method listed is relevant to your study. If you are not sure if a list item applies to your research, read the appropriate section before selecting a response.

### Materials & experimental systems

| n/a | Involved in the study |
|---|---|
| ☒ ☐ | Antibodies |
| ☒ ☐ | Eukaryotic cell lines |
| ☒ ☐ | Palaeontology and archaeology |
| ☒ ☐ | Animals and other organisms |
| ☒ ☐ | Clinical data |
| ☒ ☐ | Dual use research of concern |
| ☒ ☐ | Plants |

### Methods

| n/a | Involved in the study |
|---|---|
| ☒ ☐ | ChIP-seq |
| ☒ ☐ | Flow cytometry |
| ☒ ☐ | MRI-based neuroimaging |

# Plants

Seed stocks

NA

Novel plant genotypes

NA

Authentication

NA

