## [Peer Review File · Nature Ecology & Evolution]

Peer Review Information

Journal: Nature Ecology & Evolution

Manuscript Title: Biodiversity-production feedback effects lead to intensification traps in agricultural landscapes

Corresponding author name(s): Alfred Burian

Editorial Notes:

Reviewer Comments & Decisions:

Decision Letter, initial version:

3rd July 2023

Dear Dr Burian,

Your manuscript entitled "Biodiversity-production feedback effects lead to intensification traps in agricultural landscapes" has now been seen by three reviewers, whose comments are attached. The reviewers have raised a number of concerns which will need to be addressed before we can offer publication in Nature Ecology & Evolution. We will therefore need to see your responses to the criticisms raised and to some editorial concerns, along with a revised manuscript, before we can reach a final decision regarding publication.

We therefore invite you to revise your manuscript taking into account all reviewer and editor comments. Please highlight all changes in the manuscript text file.

* If you have not done so already please begin to revise your manuscript so that it conforms to our Article format instructions at <http://www.nature.com/natecolevol/info/final-submission>. Refer also to any guidelines provided in this letter.

2[REDACTED]

Nature Ecology & Evolution is committed to improving transparency in authorship. As part of our efforts in this direction, we are now requesting that all authors identified as 'corresponding author' on published papers create and link their Open Researcher and Contributor Identifier (ORCID) with their account on the Manuscript Tracking System (MTS), prior to acceptance. ORCID helps the scientific community achieve unambiguous attribution of all scholarly contributions. You can create and link your ORCID from the home page of the MTS by clicking on 'Modify my Springer Nature account'. For more information please visit www.springernature.com/orcid.

[REDACTED]

Reviewer expertise:

Reviewer #1: agricultural landscape conservation/restoration

Reviewer #2: agrobiodiversity. See also marked up manuscript attachment

Reviewer #3: Agricultural tipping points

Reviewers' comments:

Reviewer #1 (Remarks to the Author):

This paper explores the conditions (relationships between biodiversity, yield, and working land area) that create differential risks of intensification traps using simulation-based models. Impressively, the study integrates case studies that represent archetypes of ag systems and does a literature review to refine the model parameter values and uncertainty around those values. The results are compelling

2and broadly interesting and will likely be motivating to researchers across systems thinking about risk of intensification traps - where they are realized (and where they are not) and how well that aligns with the model (the authors explore this a bit, but also really motivate future work in this space!)

While the paper is well documented (both in the text, and with a thorough code repository) my main concern is regarding the interpretation of some of the results. The paper would benefit from more discussion around where the results are simply a reflection of a given set of model assumptions and where they interesting emergent properties of the interacting parts of the system (of course, this is a blurry line). I found the figures to be compelling because they focus very clearly on the parameter spaces where there is high risk for intensification traps. However, the discussion/results did not as effectively convey this idea in its current form. For example, in the discussion the authors say "Such small-loss large-win trade-offs are exemplified in the rice case study where a reduction of maximal crop production by 5% resulted in a doubling of biodiversity (Fig. 3)." Is this not just by definition of your equation? What about under a different relationship curves/parameter values explored? I don't understand how this is a finding, particularly the exact percentage given the sensitivity of these findings to shifts in model structure and parameters. Overall, I think the results/discussion section would be greatly strengthened by navigating the parameter spaces that drive certain outcomes — when and where might we see traps, not what a specific model run gave as an outcome.

Another related comment is related to the archetypes. It would be helpful if this was more readily explained in the main text and explored throughout the discussion. Where would you expect to see a given result given what we know about the parameter spaces that drive those results? The archetypes are helpful, but it doesn't, in its current form, feel clear what other archetypes might exist (my understanding is that you don't archetype the entire parameter/model space, rather just provide three case examples).

A final comment is with regards to the use of the term "opportunity cost" in the context of production loss. The authors define the term a couple of times thought out (e.g., line 180 as "potential production losses resulting from reduced management intensities (further referred to as opportunity costs)") however, I think this is a bit misleading - is there a reason to not just refer to it as "production loss" (which is distinct from the common definition of opportunity cost and therefore confusing). If there is a good reason for using opportunity cost rather than production loss, please make it clear in the text.

A few super minor comments:

Abstract (line 34): clarify what you mean by "integrating biodiversity" here — e.g., "considering the feedbacks between biodiversity and production"

Line 36: Reviews? Are there multiple lit reviews included?

a positive note: really enjoyed the explanation of the intensity traps in the introduction — authors made it super clear what they meant (and did not mean) by that, which made the whole paper more clear and I think is important given the likely readership!

Figures 3-4 : can you label the panels (A,B,C,...) and reflect that in the captions — its quite hard to

3follow exactly which panel you are referencing in some cases. Overall the visualizations are super effective

Reviewer #2 (Remarks to the Author):

I congratulate the authors of this manuscript ("Biodiversity-production feedback effects lead to intensification traps in agricultural landscapes") to this important contribution to biodiversity-production developments and debates that are of high global relevance.

I find this manuscript to be very clearly written and structured - both with respect to the qualitative and quantitative approaches and arguments used throughout.

Therefore, I have only a few minor comments to add to the main text (i.e. mainly suggestions to discuss the relevance and potential implications of (a) considering alternative biodiversity measures than species richness; and (b) land use history); as well as to the figures (i.e. mainly related to font sizes). All my comments are included in the attached source file.

This manuscript certainly presents a highly relevant contribution to identify and prevent potential intensification traps in agriculture. I hope the authors find my suggestions helpful and look forward to seeing this work published and discussed across diverse disciplines and perspectives.

Reviewer #3 (Remarks to the Author):

This paper sets out to investigate an interesting and important topic, whether agricultural intensification leads to "intensification traps", where agricultural intensification leads to reduced biodiversity, which in turn leads to reduced yield, by way of lower delivery of key ecosystem services. In general, the paper is well written, although a bit technical in some places. However, the exact aim of the paper is a bit unclear to me. Is the main aim to describe a framework to detect potential intensification traps? Or is it to test how widespread such intensification traps are in reality, across agricultural production systems? Now it seems to try to do a bit of both...

In either case, I have problems with the selection of case studies (i.e. the three "archetypal landscapes"). The selection per se makes sense, and they represent three major types of agricultural production systems, but since the diversity of production systems is huge, I would have preferred to see a clearer motivation why the authors select specifically these three systems, and what limitations this selection implies, in terms of interpretation of the results of the simulations. This is described to some degree in the Supplementary information, but I think it should be presented in the main text.

Specific comments:

In relationship "D", the authors seem to imply that the only effects of agricultural intensification on biodiversity act through input of fertilizers and pesticides. But intensification is much more multi-faceted than this, and includes crop rotations, frequency and magnitude of tillage and other mechanical

4actions, irrigation etc. How is this accounted for in the model(s)

Line 37: Where does the 73% come from? I tried to find this in the Results and Discussion section, but could not find it there.

Line 37-38: "major calorie production systems" – Please explain what this means

Line 48-49: Please explain why you chose the term "conventional intensification". To me, this only makes sense in contrast to ecological intensification. So perhaps one or a few sentences on the contrast between "conventional" and ecological intensification here.

Line 50-51: "...can result in strong negative feedback effects on yield...": Please explain here why this is the case, i.e. that the yield is (at least partly) dependent on biodiversity through ecosystem functions. This is explained later, but when you only read this paragraph it is not clear to the reader.

Line 66 ff: "Further, yield declines due to loss in biodiversity can feedback to management and result in additional intensification...": To me, this would be the definition of an intensification trap (i.e. not the "lose-lose" situation that the authors use as the definition.

Line 303 ff: It is of course fine to exclude ecological intensification here, but it would still be interesting to assess, or at least discuss (!) if also ecological intensification might lead to intensification traps.

Line 326-327: Check formatting of these references.

In supplementary information

Line 195 ff: "the forest hundred returns" – But this depends on how they were sorted, right? Please clarify!

Table S3: - Where only pollination and pest control considered as important ecosystem service in Relationship C? And were only animal biodiversity considered in Relationship D? How do these limitations influence the overall results?

*****END*****

Author Rebuttal to Initial comments

Reviewer #1

This paper explores the conditions (relationships between biodiversity, yield, and working land area) that create differential risks of intensification traps using simulation-based models. Impressively, the study integrates case studies that represent archetypes of ag systems and does a literature review to refine the model parameter values and uncertainty around those values. The results are compelling and broadly interesting and will likely be motivating to researchers across systems thinking about risk of

5intensification traps - where they are realized (and where they are not) and how well that aligns with the model (the authors explore this a bit, but also really motivate future work in this space!)

Response: Thank you for the time and efforts invested in the very helpful review of our work. We agree with all of the major points raised by the reviewer and have revised our manuscript accordingly. The reviewer's suggestions primarily addressed framing and structuring of our manuscript and hence, we are confident that the revised version has gained substantially in clarity thanks to the reviewer's inputs.

While the paper is well documented (both in the text, and with a thorough code repository) my main concern is regarding the interpretation of some of the results. The paper would benefit from more discussion around where the results are simply a reflection of a given set of model assumptions and where they interesting emergent properties of the interacting parts of the system (of course, this is a blurry line).

Response: Thank for this very helpful recommendation – in essence, our model was not predetermined to lead to intensification traps, but they rather emerged (or not) based on the combination of chosen model constants. The specification of model constants in two of our three analyses (generation of artificial landscapes and evaluation of archetypal landscapes) was based on our systematic literature reviews. Hence, the main results of these analyses represent emergent properties conditioned by parameters identified from our literature review, i.e. reflecting real-world conditions and their variability. We have now added this important clarification at the beginning of our result and discussion section (Lines 156-158 and 173-175).

Of course, our model results are certainly not independent of the chosen model structure. We now make sure to discuss this consideration to a greater extent in later sections of the result and discussion section (see e.g. Line 258-263 and subsequent paragraphs) and in an extra section in the supplementary information that has now been created for this purpose (Section A3).

I found the figures to be compelling because they focus very clearly on the parameter spaces where there is high risk for intensification traps. However, the discussion/results did not as effectively convey this idea in its current form. For example, in the discussion the authors say “Such small-loss large-win trade-offs are exemplified in the rice case study where a reduction of maximal crop production by 5% resulted in a doubling of biodiversity (Fig. 3).” Is this not just by definition of your equation? What about under a different relationship curves/parameter values explored? I don't understand how this is a finding, particularly the exact percentage given the sensitivity of these findings to shifts in model structure and parameters.

Response: The reviewer is absolutely correct – this specific result reflects the baseline parametrisation of our model equation and although we carefully document our parametrisation in the supplementary

information, it should not be overstated. However, our main result here (i.e. that a large gain in biodiversity can be obtained with just a small loss of crop production due to the non-linear pattern of opportunity-cost-curves) is consistently found in all three of case-study and in the vast majority of our artificial landscape types. It hence represents an emergent property that is largely independent of specific parametrisation. We now substantially reworked this section to clarify this important point (Lines 184-190).

Overall, I think the results/discussion section would be greatly strengthened by navigating the parameter spaces that drive certain outcomes — when and where might we see traps, not what a specific model run gave as an outcome.

Response: We agree with the reviewer – the exploration of the parameter space is a very important exercise that can greatly increase our understanding of the underlying drivers. Our intention was that our sensitivity analysis and the section that is now titled ‘What increases the risk of intensification traps?’ (Line 191) fulfils this purpose. We now clarified the entry statement of this section and further tried to remove technicalities from the text. Hence, the exploration of the model’s parameter space and the identification of where traps likely occur is now a substantial element of our analysis (~40% of the result and discussion section with the main findings being highlighted in the conclusion section – Lines 313-318). Additionally, we have taken care now to not overstate individual model runs and justify our conclusions based on multiple lines of evidence.

Another related comment is related to the archetypes. It would be helpful if this was more readily explained in the main text and explored throughout the discussion. Where would you expect to see a given result given what we know about the parameter spaces that drive those results? The archetypes are helpful, but it doesn’t, in its current form, feel clear what other archetypes might exist (my understanding is that you don’t archetype the entire parameter/model space, rather just provide three case examples).

Response: Thanks for your comment. We have now improved this section by stating clearly (i) that we choose only three out of many possible archetypal landscapes and (ii) explaining the reasoning for our choice. The reviewer is absolutely correct, the chosen case-studies can, of course, never represent the entire parameter space. However, we think that they, especially after the now-implemented clarifications, complement more systematic assessment (i.e., our sensitivity analysis in the ‘What drives the risk of trap situations’ section) while also helping to contextualise our results.

A final comment is with regards to the use of the term “opportunity cost” in the context of production loss. The authors define the term a couple of times thought out (e.g., line 180 as “potential production losses resulting from reduced management intensities (further referred to as opportunity costs)”).

however, I think this is a bit misleading - is there a reason to not just refer to it as “production loss” (which is distinct from the common definition of opportunity cost and therefore confusing). If there is a good reason for using opportunity cost rather than production loss, please make it clear in the text.

Response: Thank you for this comment. Upon reflection, we fully agree with reviewer. In contrast to the opportunity-cost curves, which truly reflect opportunity costs, the term had not been correctly used in the context of this section. We made according changes to wording of the section.

A few super minor comments:

Abstract (line 34): clarify what you mean by “integrating biodiversity” here — e.g., “considering the feedbacks between biodiversity and production”

Response: We followed the advice and changed to wording to ‘biodiversity feedback’.

Line 36: Reviews? Are there multiple lit reviews included?

Response: We changed the wording to systematic literature reviews and we have clarified at the end of the introduction indeed one systematic review was implemented for each model constant that needed to be parameterised. This information was previously provided in the SI, but we agree that it needs to be summarised in the main text as well (now in Lines 142-146).

a positive note: really enjoyed the explanation of the intensity traps in the introduction — authors made it super clear what they meant (and did not mean) by that, which made the whole paper more clear and I think is important given the likely readership!

Response: Thank you very much!

Figures 3-4: can you label the panels (A,B,C,...) and reflect that in the captions — its quite hard to follow exactly which panel you are referencing in some cases. Overall the visualizations are super effective

Response: We followed the advice and added panel labels to figures and increased the clarity of the figure legend of figure 3.

Reviewer #2

I congratulate the authors of this manuscript ("Biodiversity-production feedback effects lead to intensification traps in agricultural landscapes") to this important contribution to biodiversity-production developments and debates that are of high global relevance. I find this manuscript to be very clearly written and structured - both with respect to the qualitative and quantitative approaches and arguments used throughout.

Response: Thank you, we are very happy to hear that our work resonated with you and thank you for the time and efforts invested!

Therefore, I have only a few minor comments to add to the main text (i.e. mainly suggestions to discuss the relevance and potential implications of (a) considering alternative biodiversity measures than species richness; and (b) land use history; as well as to the figures (i.e. mainly related to font sizes). All my comments are included in the attached source file.

Response: We copied the specific comments that were directly included into the manuscript in the selection below to provide detailed replies. The comments were much appreciated and we followed most of them if not word count considerations (which the reviewer acknowledges her/himself) made us do otherwise.

Specific comments:

Line 53: I suggest to split this statement into two sentences.

Response: We followed the advice.

Line 57: I suggest to add this term (tipping points) to the definitions in Table 1

Response: The term was only mentioned in our manuscript once. As mentioned above, we try to manage the word count of the manuscript and we believe that the readers of our article will have a general idea of the meaning of the term. We believe that this suffices in this context, but we are happy to remove the term if the review thinks that it leads to unnecessary confusion.

Line 58: I suggest to specify these with some examples in brackets (e.g., pollination, pest suppression, etc.).

Response: We followed the advice and added specific examples.

Line 60: Also here, I suggest to add some specific examples in brackets for improved clarity (e.g., pollination and pest suppression services contributing to increased yield quantity and quality).

Response: We modified the structure of the sentence to make it more intuitively understandable (Line 60 of the revised manuscript).

Line 76: I suggest to both briefly define this term here (to reach a larger audience) and add it (its definition) to Table 1.

Response: We followed the advice and included the term now in the definition table.

Line 79: Please indicate the scale here – is this a global assessment?

Response: Thank you for pointing this out – clarifying the scale of our analysis was an important addition to our ms. We have done so now in Lines 151 and 327 of the revised manuscript.

Line 84: As you introduce these as being “5”, I suggest to number the relationships below (instead of using letters A-E).

Response: Thank you for the suggestion. However, since we also refer to first order and second order indirect relationships (see Fig.1), we chose to use letters to avoid a confusion among terms.

Line 85: It would be great to add a statement on the relevance of other biodiversity indicators/measures such as abundance, phylogenetic and functional diversity to be considered in such assessments, frameworks and their implementation – together with some relevant references, for example:

<https://www.science.org/doi/10.1126/sciadv.aax0121>

<https://www.pnas.org/doi/abs/10.1073/pnas.1701370114>

<https://besjournals.onlinelibrary.wiley.com/doi/full/10.1111/1365-2664.13970>

Response: We fully agree with this suggestion. We now address potential complementary effects of other biodiversity indices briefly in the discussion (Line 258-261, revised ms version) and added a section in the supplementary information (Section A3) to discuss potential ramifications more thoroughly. We included the references (thank you) which we found to be useful complementary citations.

Line 90: I suggest to add a specification as for (i) here.

Response: We followed the recommendation and added a specification.

Line 93: Please consider adding a statement on the importance of land use history in the overall context /conclusions of your work – together with relevant references, for example:

<https://www.pnas.org/doi/abs/10.1073/pnas.1910023117>

<https://conbio.onlinelibrary.wiley.com/doi/full/10.1111/conl.12740>

Response: Yes, we agree that land-use history is of large importance in the context of intensification traps. We state this now specifically in Line 66 and 292-293 of the revised manuscript and also incorporated references to support this statement!

Line 94: Reference should be a superscripted number?

Response: Thank you, we corrected that reference.

Line 95-96: Is there a reference to this statement? On the other hand, it could be expected that agricultural expansion is higher in areas that are more accessible, fertile, and politically stable than others (?). It would be interesting to provide a bit more detail on this topic /questions here and/or in the discussion.

Response: Yes, there is good evidence from e.g. Sub-Saharan Africa, which is, together with Latin America, the continent with the fastest growth in crop production area due to its rapid population growth. We now state respective references in the text.

Further, we focused in our analyses on the landscape scale and applied normalizations that prevented us from comparing high productivity and low productivity landscapes (see SI, section A3 – we added a more detailed discussion there). Hence, the question, whether expansion is higher in areas that are more accessible and fertile is indeed interesting, but beyond the scope of our study.

Line 101: If there is space, it would be good to mention main drivers here (e.g., effects on soils and ecosystem services of above-ground biodiversity; effects of decreased landscape connectivity; etc.)

Response: We followed the advice.

Line 106: This is a very optional suggestion to consider, but I prefer to use the term “suppression” instead of “control” as it reflects the mechanism (and what can be expected from it) much better in my opinion.

Response: Upon reflection, we agree with the suggestion and changed the term accordingly.

Line109-112: Are there references available to these statements?

Response: There are references demonstrating that more specialized predators are less effective in controlling diverse prey communities (<https://www.journals.uchicago.edu/doi/full/10.1086/428300>). We added the reference.

Line 117 and line 121: In this context, it seems relevant to mention /discuss the (potential) impacts of land use history on biodiversity and its responses to land use expansion and intensification. Response: We have followed the reviewer's comment and stated the importance of land-use history in the introduction. We also mention the impact of past land-uses (e.g. pesticide exposure) in the discussion as one factor influencing regional species pools (Lines 66 and 292-293).

Line 135-137: Here (and also earlier; please see my question on the spatial scale of your assessment), it would be good to provide more information on the number /diversity of assessed crop types and geographic regions (and later discuss all relevant biases and their impact on our understanding of intensification traps in the discussion and conclusions)

Response: We have added a reference to the scale of our analysis.

Line 142: A very nice and clear results/discussion and also conclusion section. Please consider my other comments in the manuscript to add clarity on some crucial considerations such as the importance of other biodiversity measures than species richness and the (potential) impacts of land use history on your suggested approaches and implications.

Response: Thank you for the complement! We followed the reviewer's earlier recommendations suggesting further clarifications.

Comments on figures: Modification of font sizes

Response: We appreciate and implemented the suggestions.

Comments on abstract:

Response: Thank you for the suggestions, we have now thoroughly revised the abstract to increase clarity and its flow whilst also adhering to word counts.

This manuscript certainly presents a highly relevant contribution to identify and prevent potential intensification traps in agriculture. I hope the authors find my suggestions helpful and look forward to seeing this work published and discussed across diverse disciplines and perspectives.

Response: Thank you, we are glad that our work found your approval and hope indeed that it can indeed contribute to the discussion on land-use, agricultural production, and biodiversity loss.

Reviewer #3

This paper sets out to investigate an interesting and important topic, whether agricultural intensification leads to "intensification traps", where agricultural intensification leads to reduced biodiversity, which in turn leads to reduced yield, by way of lower delivery of key ecosystem services.

11In general, the paper is well written, although a bit technical in some places. However, the exact aim of the paper is a bit unclear to me. Is the main aim to describe a framework to detect potential intensification traps? Or is it to test how widespread such intensification traps are in reality, across agricultural production systems? Now it seems to try to do a bit of both...

Response: Thank you this comment, also reviewer 1 pointed out that our manuscript required some additional clarifications and hence this was the main focus of our revisions.

Specifically, we:

- (i) Clarified the section stating the aim of our manuscript. We now clearly state that our aim is both to refine previous frameworks describing intensification traps, which included the development of a quantitative modelling framework, and to illustrate the utility of such a model in real world situations.
- (ii) Reduced technical language, especially in the section 'What drives the risk of trap situations?'. While the level of details was positively commented on by other reviewers, we now made the text more easily accessible.
- (iii) Provided a clear justification for the choice of our archetypal case-study and their inclusion in our analysis (see below).
- (iv) Clearly stated which are emergent properties from our analysis (for details, see responses to reviewer 1).
- (v) Improved on the discussion of model assumptions (see below).

In either case, I have problems with the selection of case studies (i.e. the three "archetypal landscapes"). The selection per se makes sense, and they represent three major types of agricultural production systems, but since the diversity of production systems is huge, I would have preferred to see a clearer motivation why the authors select specifically these three systems, and what limitations this selection implies, in terms of interpretation of the results of the simulations. This is described to some degree in the Supplementary information, but I think it should be presented in the main text.

Response: Thank you, this was an important comment to improve the clarity of our manuscript. We followed the advice and added a motivation for the selection of archetypal case studies when they are first mentioned in the results (Lines 165-167). We also clearly state now that the main purpose of the case-studies was to contextualise our results, but we only deduce key message when the patterns observed in case-studies concord with the results of other analyses we performed (Lines 183-190).

Specific comments:

In relationship "D", the authors seem to imply that the only effects of agricultural intensification on biodiversity act through input of fertilizers and pesticides. But intensification is much more multi-faceted than this, and includes crop rotations, frequency and magnitude of tillage and other mechanical actions, irrigation etc. How is this accounted for in the model(s)

Response: We fully agree, conventional intensification encompasses many different practices and reducing it to only fertiliser and pesticide inputs (which we did in our model) is a substantial simplification. The same is true for our representation of biodiversity (i.e. we only consider richness whereas biodiversity is likewise multidimensional and also evenness or genetic redundancy, etc. can have important complementary effects). Yet, including interlinkages between these multiple dimensions would substantially increase model complexity. We now clearly state this in the result and discussion section (Line 257-261) and address e.g. differences between conventional and ecological intensification in the subsequent paragraphs and the methods section (Lines 276-282 and 329-335). Additionally, a new section of the SI (Section A3) that was added provides a better overview of the most important model assumptions. Overall, we attempt to carefully balance clarity of our models with their realism, resulting naturally in a compromise.

Line 37: Where does the 73% come from? I tried to find this in the Results and Discussion section, but could not find it there.

Response: Thank you for pointing this out – the value states the proportion of landscape types in our analysis of artificial landscapes where intensification traps emerge at highest management intensities. We have added this important context information to the results and discussion section (155-158).

Line 37-38: “major calorie production systems” – Please explain what this means

Response: We replaced calorie with cereal to use a more specific term.

Line 48-49: Please explain why you chose the term “conventional intensification”. To me, this only makes sense in contrast to ecological intensification. So perhaps one or a few sentences on the contrast between “conventional” and ecological intensification here.

Response: We agree – the term primarily makes sense contrasting conventional with ecological intensification. This contrast is now also clarified in the definition table and our focus on conventional intensification is addressed in the result and discussion section as well as in the method section (Lines 276-282 and 329-335).

Line 50-51: “...can result in strong negative feedback effects on yield...”: Please explain here why this is the case, i.e. that the yield is (at least partly) dependent on biodiversity through ecosystem functions. This is explained later, but when you only read this paragraph it is not clear to the reader.

Response: We followed the suggestion and added the loss of ecosystem functions here. In the next paragraph we specify now what these ecosystem services are.

Line 66 ff: “Further, yield declines due to loss in biodiversity can feedback to management and result in additional intensification...”: To me, this would be the definition of an intensification trap (i.e. not the “lose-lose” situation that the authors use as the definition.

Response: We see the argument of the reviewer that, in analogy with poverty traps, a self-reinforcing element is required to create a trap situation. However, we apply a different definition of trap situations. In our opinion, traps emerge when barriers exist that prevent a change in management practices. For example, natural communities will require time to recover after a switch to less intensive agricultural practices. However, if these less intensive agricultural practices rely heavily on biodiversity, production will be very low until required levels of biodiversity have been restored. The associated production losses in the near term represent such a barrier and create a trap situation. The trap will be of course even more severe (and harder to escape from) if it is self-reinforcing. We have clarified this now in the text (Lines 70-73).

Line 303 ff: It is of course fine to exclude ecological intensification here, but it would still be interesting to assess, or at least discuss (!) if also ecological intensification might lead to intensification traps.

Response: We have revised this section and now clarify that the inclusions of semi-natural landscape elements represents one agroecological/diversification practice and hence this form of ecological intensification is included explicitly in our framework. However, we also clarified that we did not include the many other practices associated to ecological intensification into our model.

It is an interesting point that also ecological intensification might lead to intensification traps. This will primarily depend on applied definitions, but we do think that this can happen in very specific cases. However, as ecological intensification is not a main focus of our analysis, we think that a more specific discussion of this point would be too much of a side-step. However, we did extend our discussion on ecological intensification in general (276-284).

Line 326-327: Check formatting of these references.

Response: Thanks, references were revised.

In supplementary information

Line 195 ff: “the forest hundred returns” – But this depends on how they were sorted, right? Please clarify!

Response: We clarified accordingly.

Table S3: - Where only pollination and pest control considered as important ecosystem service in Relationship C? And were only animal biodiversity considered in Relationship D? How do these limitations influence the overall results?

Response: Yes, we included only pollination and natural pest control for relationship C - see above for a detailed explanation for this choice and e.g. section A3 in the SI for an in-depth discussion. Data underlying relationship C included many different taxonomic groups such as plants, pollinators, birds and parasitoid wasps (see Appendix 1).

Decision Letter, first revision:

23rd November 2023

Dear Dr. Burian,

Thank you for submitting your revised manuscript "Biodiversity-production feedback effects lead to intensification traps in agricultural landscapes" (NATECOLEVOL-23051050A). It has now been seen again by the original reviewers and their comments are below. The reviewers find that the paper has improved in revision, and therefore we'll be happy in principle to publish it in Nature Ecology & Evolution, pending minor revisions to satisfy the reviewers' final requests and to comply with our editorial and formatting guidelines.

[REDACTED]

Reviewer #1 (Remarks to the Author):

The authors sufficiently addressed my comments, and I would suggest the paper for publication.

Reviewer #2 (Remarks to the Author):

I would like to thank the authors for considering my comments, all of which were clearly taken into account. The revised manuscript is, in my opinion, an excellent contribution to this journal and I have no further comments. I wish the authors every success with the publication and promotion of their manuscript and look forward to seeing it published!

Reviewer #3 (Remarks to the Author):

15The authors have made a great job in incorporating comments from me and from the other reviewers in this revised version of the manuscript.

I only have a few additional minor comments:

Line 32: Just "intensification" rather than "conventional intensification" might be better here?

Line 37: 73% of what? (of landscape types, I assume, but this could be clarified)

Introduction, first paragraph: I still think that Ecological intensification (as contrast to conventional intensification) should be mentioned in the introduction, or as a separate entry in Table 1. Now it appears first in the Methods. This would clarify the context for readers who are not familiar with this concept

Our ref: NATECOLEVOL-23051050A

8th December 2023

Dear Dr. Burian,

Thank you for your patience as we've prepared the guidelines for final submission of your Nature Ecology & Evolution manuscript, "Biodiversity-production feedback effects lead to intensification traps in agricultural landscapes" (NATECOLEVOL-23051050A). Please carefully follow the step-by-step instructions provided in the attached file, and add a response in each row of the table to indicate the changes that you have made. Please also check and comment on any additional marked-up edits we have proposed within the text. Ensuring that each point is addressed will help to ensure that your revised manuscript can be swiftly handed over to our production team.

****We would like to start working on your revised paper, with all of the requested files and forms, as soon as possible (preferably within two weeks). Please get in contact with us immediately if you anticipate it taking more than two weeks to submit these revised files.****

If you have not done so already, please alert us to any related manuscripts from your group that are under consideration or in press at other journals, or are being written up for submission to other journals (see: <https://www.nature.com/nature-research/editorial-policies/plagiarism#policy-on->

16duplicate-publication for details).

In recognition of the time and expertise our reviewers provide to Nature Ecology & Evolution's editorial process, we would like to formally acknowledge their contribution to the external peer review of your manuscript entitled "Biodiversity-production feedback effects lead to intensification traps in agricultural landscapes". For those reviewers who give their assent, we will be publishing their names alongside the published article.

Nature Ecology & Evolution offers a Transparent Peer Review option for new original research manuscripts submitted after December 1st, 2019. As part of this initiative, we encourage our authors to support increased transparency into the peer review process by agreeing to have the reviewer comments, author rebuttal letters, and editorial decision letters published as a Supplementary item. When you submit your final files please clearly state in your cover letter whether or not you would like to participate in this initiative. Please note that failure to state your preference will result in delays in accepting your manuscript for publication.

Cover suggestions

We welcome submissions of artwork for consideration for our cover. For more information, please see our [guide for cover artwork](https://www.nature.com/documents/Nature_covers_author_guide.pdf).

Nature Ecology & Evolution has now transitioned to a unified Rights Collection system which will allow our Author Services team to quickly and easily collect the rights and permissions required to publish your work. Approximately 10 days after your paper is formally accepted, you will receive an email in providing you with a link to complete the grant of rights. If your paper is eligible for Open Access, our Author Services team will also be in touch regarding any additional information that may be required to arrange payment for your article.

Please note that *Nature Ecology & Evolution* is a Transformative Journal (TJ). Authors may publish their research with us through the traditional subscription access route or make their paper immediately open access through payment of an article-processing charge (APC). Authors will not be required to make a final decision about access to their article until it has been accepted. [Find out more about Transformative Journals](https://www.springernature.com/gp/open-research/transformative-journals)

Authors may need to take specific actions to achieve [17](https://www.springernature.com/gp/open-research/funding/policy-compliance- compliance with funder and institutional open access mandates. If your research is supported by a funder that requires immediate open access (e.g. according to Plan S principles) then you should select the gold OA route, and we will direct you to the compliant route where possible. For authors selecting the subscription publication route, the journal's standard licensing terms will need to be accepted, including https://www.nature.com/nature-portfolio/editorial-policies/self-archiving-and-license-to-publish. Those licensing terms will supersede any other terms that the author or any third party may assert apply to any version of the manuscript.

For information regarding our different publishing models please see our Transformative Journals page. If you have any questions about costs, Open Access requirements, or our legal forms, please contact ASJournals@springernature.com.

[REDACTED]

[REDACTED]

Reviewer #1:

Remarks to the Author:

The authors sufficiently addressed my comments, and I would suggest the paper for publication.

Reviewer #2:

Remarks to the Author:

I would like to thank the authors for considering my comments, all of which were clearly taken into account. The revised manuscript is, in my opinion, an excellent contribution to this journal and I have no further comments. I wish the authors every success with the publication and promotion of their manuscript and look forward to seeing it published!

Reviewer #3:

Remarks to the Author:

The authors have made a great job in incorporating comments from me and from the other reviewers in this revised version of the manuscript.

I only have a few additional minor comments:

18Line 32: Just "intensification" rather than "conventional intensification" might be better here?

Line 37: 73% of what? (of landscape types, I assume, but this could be clarified)

Introduction, first paragraph: I still think that Ecological intensification (as contrast to conventional intensification) should be mentioned in the introduction, or as a separate entry in Table 1. Now it appears first in the Methods. This would clarify the context for readers who are not familiar with this concept

Final Decision Letter:

26th January 2024

Dear Dr Burian,

We are pleased to inform you that your Article entitled "Biodiversity-production feedback effects lead to intensification traps in agricultural landscapes", has now been accepted for publication in Nature Ecology & Evolution.

Over the next few weeks, your paper will be copyedited to ensure that it conforms to Nature Ecology and Evolution style. Once your paper is typeset, you will receive an email with a link to choose the appropriate publishing options for your paper and our Author Services team will be in touch regarding any additional information that may be required

Due to the importance of these deadlines, we ask you please us know now whether you will be difficult to contact over the next month. If this is the case, we ask you provide us with the contact information (email, phone and fax) of someone who will be able to check the proofs on your behalf, and who will be available to address any last-minute problems . Once your paper has been scheduled for online publication, the Nature press office will be in touch to confirm the details.

Acceptance of your manuscript is conditional on all authors' agreement with our publication policies (see www.nature.com/authors/policies/index.html). In particular your manuscript must not be published elsewhere and there must be no announcement of the work to any media outlet until the publication date (the day on which it is uploaded onto our web site).

Please note that *Nature Ecology & Evolution* is a Transformative Journal (TJ). Authors may publish their research with us through the traditional subscription access route or make their paper immediately open access through payment of an article-processing charge (APC). Authors will not be required to make a final decision about access to their article until it has been accepted. [Find out more about Transformative Journals](https://www.springernature.com/gp/open-research/transformative-journals)

19Authors may need to take specific actions to achieve [compliance](https://www.springernature.com/gp/open-research/funding/policy-compliance-faqs) with funder and institutional open access mandates. If your research is supported by a funder that requires immediate open access (e.g. according to [Plan S principles](https://www.springernature.com/gp/open-research/plan-s-compliance)) then you should select the gold OA route, and we will direct you to the compliant route where possible. For authors selecting the subscription publication route, the journal's standard licensing terms will need to be accepted, including <https://www.nature.com/nature-portfolio/editorial-policies/self-archiving-and-license-to-publish>. Those licensing terms will supersede any other terms that the author or any third party may assert apply to any version of the manuscript.

We welcome the submission of potential cover material (including a short caption of around 40 words) related to your manuscript; suggestions should be sent to Nature Ecology & Evolution as electronic files (the image should be 300 dpi at 210 x 297 mm in either TIFF or JPEG format). Please note that such pictures should be selected more for their aesthetic appeal than for their scientific content, and that colour images work better than black and white or grayscale images. Please do not try to design a cover with the Nature Ecology & Evolution logo etc., and please do not submit composites of images related to your work. I am sure you will understand that we cannot make any promise as to whether any of your suggestions might be selected for the cover of the journal.

You can generate the link yourself when you receive your article DOI by entering it here: <http://authors.springernature.com/share>.

[REDACTED]

P.S. Click on the following link if you would like to recommend Nature Ecology & Evolution to your librarian <http://www.nature.com/subscriptions/recommend.html#forms>

** Visit the Springer Nature Editorial and Publishing website at http://editorial-jobs.springernature.com?utm_source=ejp_NEcoE_email&utm_medium=ejp_NEcoE_email&utm_campaign=ejp_NEcoE for more information about our career opportunities. If you have any questions please click [here](mailto:editorial.publishing.jobs@springernature.com).**